

# Eddy covariance carbonyl sulphide flux measurements with a quantum cascade laser absorption spectrometer

Katharina Gerdel[1], Felix M. Spielmann[1], Albin Hammerle[1], Georg Wohlfahrt[1]

[1]Institut of Ecology, University of Innsbruck, Innsbruck, Austria

*Correspondence to*: Georg Wohlfahrt (georg.wohlfahrt@uibk.ac.at)

**Abstract.** The trace gas carbonyl sulphide (COS) has lately received growing interest in the eddy covariance (EC) community due to its potential to serve as an independent approach for constraining gross primary production and canopy stomatal conductance. Thanks to recent developments of fast-response high-precision trace gas analysers (e.g. quantum cascade laser absorption spectrometers (QCLAS)), a handful of EC COS flux measurements have been published since 2013.

To date, however, a thorough methodological characterisation of QCLAS with regard to the requirements of the EC technique and the necessary processing steps has not been conducted. The objective of this study is to provide for the first time a rigorous analysis of the most widely used QCLAS model for making defensible EC COS flux measurements. Data were collected from May to October 2015 at a temperate mountain grassland in Tyrol, Austria. Analysis of the Allan variance of high-frequency concentration measurements revealed laser drift to occur under field conditions after an

averaging time of around 50 s. We thus explored the use of two high-pass filtering approaches (linear detrending and recursive filtering) as opposed to block averaging for covariance computation. Spectral analysis revealed considerable noise at higher frequencies, appearing to influence the high-frequency region of co-spectra (and therefore covariances). By applying a finite-impulse response filter we removed the noise-affected spectral region. The effects of this digital high- and low-pass filtering, and additional low-pass filtering due to the eddy covariance system design, were corrected for using a

site-specific reference model co-spectrum and a series of transfer functions. An independent validation of these post-processing steps was achieved by comparison of the $CO_2$ and $H_2O$ flux measurements obtained with the QCLAS against those obtained with a closed-path infrared gas analyser. While the validation showed good correspondence and minor statistical differences between the three different high-pass filtering approaches - the benefits of high pass filtering clearly emerged as a reduction of the random flux uncertainty and a higher fraction of data passing the applied QA/QC criterions.

We conclude that the most widely used QCLAS can be used to make defensible COS flux measurements provided the appropriate corrections are applied.



## 1. Introduction

The need for understanding human-induced climate change and in this context understanding the pathways and processes determining the global carbon cycle dynamics triggered the increased use of the eddy covariance (EC) method for carbon dioxide ($CO_2$) flux measurements and resulted in the establishment of the first flux measurement network, the EUROFLUX

project, in 1996. Today, the EC method is routinely used for measurements of the energy and trace gas exchange between the atmosphere and the biosphere at >500 sites within the FLUXNET project (Baldocchi, 2003, Baldocchi, 2014), mainly focusing on the exchange processes of $CO_2$ and $H_2O$.

By partitioning the biosphere-atmosphere $CO_2$ fluxes (net ecosystem exchange - NEE) into uptake (gross primary

productivity – GPP) and release (ecosystem respiration - $R_{eco}$) it is possible to quantify the two main processes underlying the NEE. To this end, so-called eddy covariance $CO_2$ flux partitioning algorithms are used which exploit the strong contrast between nighttime release and daytime net uptake of $CO_2$ (Reichstein et al., 2005, Lasslop et al., 2010), based on the extrapolation of nighttime $R_{eco}$ to daytime conditions. The fact that the theoretical models underlying the flux partitioning algorithms are highly simplistic and thus neglect or misinterpret certain processes has caused considerable criticism

(Wohlfahrt et al., 2005b, Wohlfahrt and Gu, 2015), however to date the resulting GPP and $R_{eco}$ estimates represent the major data source for data-driven assessments of the terrestrial carbon cycle (e.g. Beer et al., 2010, Mahecha et al., 2010) and the calibration of carbon cycle models (e.g. Friend et al., 2007).

Carbonyl sulphide (COS or OCS), a trace gas present in the atmosphere with an average mole fraction of 500 ppt, shares a

similar pathway during leaf uptake as $CO_2$ with the important difference that no 'respiration-like' process, i.e. emission of COS from leaves, has been reported. This co-diffusion with $CO_2$ and the one-way direction of the flux into the land biosphere (Asaf et al., 2013) have been suggested to offer the possibility of using COS as a proxy of gross primary productivity (Seibt et al., 2010, Blonquist et al., 2011). Ecosystem-atmosphere COS flux measurements were, up to very recently limited by the availability of appropriate scalar sensors with sufficient time response and sensitivity (Wohlfahrt et

al., 2012). Developments of high-precision, fast-response trace gas analyser, like e.g. quantum cascade laser spectrometers (QCLAS), propelled the analysis of COS (Stimler et al., 2010). Although these new devices are increasingly used for EC measurements (Asaf et al., 2013, Billesbach et al., 2014, Maseyk et al., 2014, Commane et al., 2015), a thorough characterisation of these instruments for flux measurements, an analysis of their limitations and the required corrections has not yet been conducted. Furthermore, previous experience with similar laser spectrometers for other trace gases such as

methane ($CH_4$) or nitrous oxide ($N_2O$) has shown that their use for EC flux measurements requires special considerations in order to result in defensible flux estimates (Kroon et al., 2007, Mammarella et al., 2010, Tuzson et al., 2010). Known issues with laser spectrometers are laser drift (Mammarella et al., 2010, Werle, 2011) and high levels of instrument noise (Smeets et al., 2009, Pierce et al., 2015). In addition, quality assurance and quality control (QA/QC) methodologies established for



infrared gas analyser-based flux measurements typically used within FLUXNET (Aubinet et al., 2012, Campbell et al., 2013, Pastorello et al., 2014), may need to be revised and adapted for flux measurements with laser spectrometers, in particular for COS, for which very few flux data are available at this point (Asaf et al., 2013, Billesbach et al., 2014, Maseyk et al., 2014, Commane et al., 2015 ).

The aim of this paper is to critically examine and validate the performance of the most widely used QCLAS for making EC COS flux measurements. To this end we use measurements obtained above a managed temperate mountain grassland in the Austrian Alps. We particularly focus on high- and low-pass filtering of the high-frequency data, adapt the QA/QC methodology, characterise the random flux uncertainty and validate our approach with independent $H_2O$ and $CO_2$ flux measurements.

## 2. Materials and Methods

### 2.1 Site description

EC flux measurements were conducted during May-Oct. 2015 at the FLUXNET site AT-Neu (http://fluxnet.ornl.gov/site/14), a managed temperate mountain grassland, located on the municipal territory of Neustift (47° 07' N, 11° 19' E) in the Stubai Valley, Austria. The site is situated at an elevation of 970 m a.s.l. in the middle of the flat bottom of the Stubai Valley with a homogenous fetch that covers about 37 ha. Dominant daytime and night time wind directions are north-east and south-west. The climate is humid continental comprising alpine influences with an average annual temperature of 6.5 °C and an average annual precipitation of 852 mm. The snow-free months usually extend from mid-March to mid-November, leaving a vegetation period of eight months. During 2015, four cutting events took place (on the 2nd of June, 7th of July, 21th of Aug. and 1st of Oct.). For information about vegetation and soil conditions we refer to Wohlfahrt et al. (2005b).

### 2.2 Eddy covariance

### 2.2.1 Data acquisition

The three wind components, as well as the speed of sound, were measured by a 3-axis sonic anemometer (R3IA, Gill Instruments, Lymington, UK) at 2.5 m above ground. Molar densities of $H_2O$ and $CO_2$ were measured by two different devices: a closed-path infra-red gas analyser (IRGA) (Li-7000, LICOR Biosciences, Lincoln, USA) and a QCLAS (Aerodyne Mini-QCL, Aerodyne Research Inc., Billerica, USA), additionally measuring COS at a wavenumber of ca. 2056 cm$^{-1}$. Raw data were acquired on two separate PCs at 20 Hz (sonic anemometer and IRGA) and 10 Hz (QCLAS) using



the Eddymeas (MPI Jena, Germany) and TDLWintel (Aerodyne, USA) software, respectively. The two PCs were synchronised in time using the NTP software (Meinberg, Germany).

The QCLAS and associated hardware (thermo cube and vacuum pump) were housed in a climate-controlled (ca. 30° C) instrument hut situated near the flux tower. We insulated the QCLAS with XPS-insulating board with openings for necessary ports for additional temperature stabilization and placed the vacuum pump on foam rubber to minimize influences by pump induced vibrations on the laser optics. The QCLAS was operated at a pressure of 20 Torr using a built-in pressure controller and temperature of the optical bench and housing was controlled to 35°C.

Sample air was drawn from the inlet through 15.7 m heated (ca. 5° above ambient) PFA Teflon tubing (4 mm inner diameter) through a filter (1-2 µm, PTFE) to the QCLAS at a flow rate of ca. 6.5 l min$^{-1}$. During the last five minutes of every half-hour, zero-air (N$_2$ 5.0, Messer, Vomp, Austria) and span gas (pressurized air, Messer, Vomp, Austria; which was cross-compared against a NOAA standard with 567 ppt COS in air) was switched into the QCLAS in order to determine the stability of the instruments zero and span. The corresponding calibration coefficients were then applied on a half-hourly basis to derive calibrated concentrations.

In order to minimise flux loss, due to vertical and longitudinal sensor separation (Massman, 2000), we installed the intake tube for the QCLAS slightly below and laterally displaced from the sonic anemometer perpendicular to the predominating wind direction.

**2.2.2 Data processing**

Subsetting of the 20 Hz sonic anemometer data to the 10 Hz resolution of the QCLAS data was done by using proprietary software (Hörtnagl et al., 2014) programmed in MATLAB 8.1.0 (R2013a, The MathWorks, Inc, USA). A 3D coordinate rotation was performed according to Kaimal and Finnigan (1994). Using the post-processing software EdiRe (University of Edinburgh), eddy fluxes of COS, CO$_2$ and H$_2$O were calculated as the covariance between the rotated vertical wind velocity and the scalar concentrations, using 25 min blocks of data. The storage flux of COS was calculated as the time rate of change in COS molar density at the reference height and was confined to ± 2.7 pmol m$^{-2}$ s$^{-1}$ in 93 % of all cases. Further details on the eddy covariance post-processing are presented in the following section.

The determination of random flux uncertainty was done following Hollinger and Richardson (2005). In short, the random uncertainty was calculated based on paired measurements at the same time of day on subsequent days with similar environmental conditions. The latter were defined as differences between pairs not exceeding the following thresholds: (i) photosynthetic photon flux density: 100 µmol m$^{-2}$ s$^{-1}$, (ii) air temperature: 3 °C, (iii) soil temperature: 2 °C, (iv) relative humidity: 20 %, and (v) wind speed: 1 m s$^{-1}$.



## 3. Results and Discussion

### 3.1 Lag-time determination

Correcting for the travel time of the gas sample in the intake tube and the resulting lag time of the scalar time series with respect to the vertical wind velocity (lag time determination), is a key post-processing step for closed-path eddy covariance

systems (McMillen, 1988). Additional time shifts may occur if the scalar and wind time series are acquired by different data acquisition systems due to drift of the respective clocks. While it would be desirable to avoid such time shifts caused by differing clocks of the data acquisition systems, we nevertheless chose this approach as preliminary tests showed that the TDLWintel software was unable to keep up the 10 Hz data acquisition rate if the sonic anemometer data were acquired on the same PC. To avoid PC clock drifts we used, based on earlier positive experience (Hörtnagl and Wohlfahrt, 2014, Pierce

et al. 2015), a software (NTP, Meinberg, Germany) which kept the clocks synchronized throughout the measurement campaign. As shown in Fig. 1, with this setup the cross correlation functions exhibited clear and, between scalars, consistent peaks (negative for COS and $CO_2$ which exhibit net uptake, positive for $H_2O$ which is released to the atmosphere). The resulting lag times were slightly longer than nominal lag times calculated based on tube flow and dimensions (1.9 s), which has been found for other closed-path eddy covariance systems as well and likely reflects unaccounted volumes (e.g. QCLAS

cell, filters), horizontal sensor separation and the scalar response time (Massman, 2000). This result illustrates the feasibility and reliability of acquiring scalar and sonic anemometer data on separate PCs provided these are appropriately synchronized.

### 3.2 High-pass filtering

Laser spectroscopy-based eddy covariance flux measurements are well known to be sensitive to laser drift, which may systematically bias flux estimates, if not corrected for (Kroon et al., 2007, Mammarella et al., 2010). In order to quantify

possible drift by the QCLAS we used so-called Allan variance plots (Werle et al., 1993, Werle, 2010), generated by feeding the QCLAS with pressurized air under ambient conditions in the instrument shelter in the field. As shown in Fig. 2, the system was dominated by white noise up to an averaging time of ca. 10 s and started to drift in an approx. linear fashion after around 50 s. Among the published COS eddy covariance flux studies (Asaf et al., 2013, Maseyk et al., 2013, Billesbach et al., 2014, Commane et al., 2015), this is the first time that QCLAS drift is characterised under field conditions. In order to

explore the effects of the drift on flux estimates, the following eddy covariance flux calculations were conducted for three high-pass filtering scenarios commonly used in the literature: (i) block averaging (BA), (ii) linear detrending (LD), and (iii) recursive filtering (RF) with a time constant of 50 s as determined from Fig. 2.

### 3.3 Low-pass filtering

Cospectral analyses, shown for COS in Fig. 3, demonstrate the expected low-pass filtering at frequencies > 0.1 Hz, caused

by the combined effects of tube attenuation, limited sensor time response, path averaging and sensor separation (e.g. Moncrieff et al., 1997, Massman, 2000). At frequencies > 0.41 Hz, however cospectra became very noisy and in many cases





rose above the sensible heat cospectrum. Inspection of the corresponding COS power spectra (Fig. 3) revealed that this deviation from the expected further attenuation of cospectral power at higher frequencies was related to the onset of noise, indicated by power spectra deviating from the theoretical inertial subrange slope and becoming independent of frequency (Eugster et al., 2007). The same behaviour was found for $CO_2$ and $H_2O$, however for these two species noise appeared at

higher frequencies ($CO_2$=2.5 Hz, $H_2O$=1.8 Hz; data not shown). Similar findings, i.e. noise appearing in the power spectra at frequencies > 0.19 Hz, were reported by Eugster et al. (2007) using a QCL for $N_2O$ flux measurements above a mixed forest in Switzerland. In contrast, Eugster et al. (2007), consistent with the idea that the EC method is a noise rejection technique for truly random noise (Wienhold et al. 1995), found no evidence for increased cospectral power caused by the high-frequency noise. In order to rule out a potential overestimation of the high-frequency flux contribution, we followed others

(e.g. Smeets et al., 2009; Pierce et al., 2015), and truncated the noise-affected region by applying a low-pass finite-impulse response (FIR) filter using the following time constants: COS=2.44 s, $CO_2$=0.41 s, $H_2O$=0.55 s. As shown in Fig. 3, the FIR filter was effective in removing unwanted (co)spectral energy in the frequencies affected by noise.

Frequency response corrections, based on a site-specific model cospectrum (Wohlfahrt et al., 2005a), were then applied

correcting for effects of (i) block averaging and high-pass filtering (where applicable) in the lower frequencies, (ii) the FIR filter in the highest frequencies, and (iii) low-pass filtering in the intermediate high frequencies. The latter correction was based on the transfer function approach by Aubinet et al. (2001) with the half-point frequency parameter ($f_0$: COS=0.22 Hz, $CO_2$=0.31 Hz, $H_2O$=0.14 Hz) being optimised against measured cospectra as shown in Fig. 3. Low-pass filtering correction factors estimated this way were lower than 1.4 (i.e. a correction of +40 %) in 89 % (COS), 96 % ($CO_2$) and 74 % ($H_2O$) of

all cases.

### 3.4 Quality control

For quality control, first, biologically or physically implausible flux values were discarded. To this end, flux ranges were determined by visually examining scatter plots of half-hourly flux data and based on this examination set to -60 to 40 pmol $m^{-2}$ $s^{-1}$ (COS), -30 to 30 µmol $m^{-2}$ $s^{-1}$ ($CO_2$) and -1 to 6 mmol $m^{-2}$ $s^{-1}$ ($H_2O$). Application of these limits removed between 2 %

and 14 % of the data, the largest removal occurring for BA, followed by LD and RF (Table 1).

Referring to the stationarity test classification of Foken and Wichura (1996) we split the data set into five categories (0-15 %, 15-30 %, 30-60 %, 60-100 %, >100 % deviation). As shown in Fig. 4, the application of high-pass filtering (LD and RF) compared to block averaging significantly increased the fraction of acceptable data ($\chi^2$-test, $p$ = 0.000 for all cases) in the

lowest category from 15 % BA to 40 % RF for COS, 32 % BA to 53 % RF for $CO_2$ and 27 % BA to 43 % RF for $H_2O$, and decreased in all other categories. The most prominent decline was observed in the 60-100 % class with 24 % BA to 12 % RF for COS, 17 % BA to 7 % RF for $CO_2$ and 26 % BA to 15 % RF for $H_2O$. Allowing a maximum deviation of 100 %, between 72 % (BA) to 84 % (RF) (COS) and 80 % (BA) to 89 % (RF) ($CO_2$) passed the stationarity test (Table 1).



In order to quantify the signal-to-noise-ratio (SNR) we adopted the approach by Wienhold et al. (1995), as described in Spirig et al. (2005). This approach involves calculation of the average covariance and its standard deviation at lag times (-180 s to -160 s and 160 s to 180 s) extended from the true lag, where the vertical wind velocity and scalar time series can be assumed to be completely de-correlated. The SNR is then expressed as the difference in covariance at the true lag and the extended lag times, divided by the standard deviation at the extended lag times. For quality control we then imposed a SNR of 1.96 $\sigma$, which would correspond to 95 % probability if data were Gaussian. As shown in Table 1, the SNR test removed the largest fraction of data for BA, followed by LD and RF and more COS and $H_2O$ fluxes compared to $CO_2$ fluxes.

Different authors have shown that during periods of low turbulence and stable stratification, typical nighttime conditions, eddy flux measurements can systematically underestimate ecosystem respiration (Goulden et al., 1996, Aubinet et al., 2000, Papale et al., 2006, Gu et al., 2005, Wohlfahrt et al., 2005a). To assess any underestimation of nighttime fluxes we plotted $CO_2$ and COS fluxes as a function of the friction velocity ($u_*$). As shown in Fig. 5, both COS and $CO_2$ fluxes were positively related to friction velocity in the lower range of friction velocities, which is presently understood to indicate insufficient turbulent mixing (Massman and Lee, 2002). Only when fluxes become independent of friction velocity, i.e. for friction velocities higher than 0.12 m s$^{-1}$ to 0.13 m s$^{-1}$ ($CO_2$) and 0.16 m s$^{-1}$ to 0.17 m s$^{-1}$ (COS), turbulent mixing is deemed sufficient to not result in a systematic underestimation of nighttime fluxes. Using an infrared gas analyser for $CO_2$ flux measurements, Wohlfahrt et al. (2005a) found $u_*$ thresholds between 0.1 m s$^{-1}$ and 0.15 m s$^{-1}$ for the same study site. Commane et al. (2015) adopted the $u_*$ threshold for $CO_2$ determined by Goulden et al. (1996) to filter COS flux measurements above Harvard forest. Using these thresholds to filter data, between 54 % - 57 % and 40 % - 43 % of the COS and $CO_2$ fluxes were excluded, respectively, with differences between the three high-pass filtering methods being minor (Table 1).

The following three quality criteria were applied independent of the scalar and high-pass filtering method:
Following McMillen (1988) we discarded all data points with the third rotation angle exceeding 10°, since larger angles lead to an insufficient definition of the third rotation angle and result in uncertainty of the output (Finnigan, 2004). As shown in Table 1, 81 % of all data passed this test.

The integral turbulence test (Foken and Wichura, 1996) is able to identify deviations of mechanical turbulence from expected Monin Obukhov similarity theory (Obukhov, 1954), caused for example by flow over complex terrain or flow distortion by measurement infrastructure. Allowing a maximum deviation of 100 %, 5 % of the data was excluded by the integral turbulence test with a larger fraction, mostly during nighttime, being excluded in the sector where the instrument hut is located.

In order to remove unwanted flux contributions by other ecosystem types (forest or settlements) beyond the study site, we



used the footprint model by Hsieh et al. (2000) and required 80 % of the flux to originate from within the grassland area, which removed 39 % of all data.

After applying all quality criteria to the data set, between 15 % (RF-COS) and 24 % (RF-$CO_2$) of the data were retained, with a larger fraction being excluded for BA as opposed to LD and RF.

**3.5 Random flux uncertainty**

The random flux uncertainty ranges of all gases decreased considerably for high pass filtering (e.g. COS: [-71.54, 73.93] (BA), [-50.85, 50.32] (LD), [-29.34, 29.99] (RF)). Figure 6 illustrates the distribution of data between the processing procedures and gases. Application of the RF high-pass filter, followed by the LD high-pass filter, reduced the number of large random uncertainties and increased the number of smaller random uncertainties, compared to the BA.

**3.6 Validation**

An independent validation of the entire post-processing chain was provided by comparison with $CO_2$ and $H_2O$ eddy covariance fluxes routinely measured with the closed-path IRGA at the study site, since the QCLAS also quantifies these two scalars. The results are shown in Fig. 7. Both data sets were filtered for flux range, stationarity, the third rotation angle, footprint and integral turbulence as described above. For $CO_2$, the results confirm that the QCLAS is capable of accurately measuring the eddy fluxes as found also by Sturm et al. (2012), who compared $CO_2$ fluxes measured with a closed-path QCLAS for the $CO_2$ isotopologues and an open-path IRGA. QCLAS-derived $H_2O$ fluxes were ca. 15 % higher than those measured with the IRGA, which may be indicative of problems with the IRGA-based $H_2O$ flux measurements (Ibrom et al., 2007) as evident in the lack of energy balance closure (the sum of the latent and sensible heat exchange falling short of the available energy by ca. 25 %; Haslwanter et al. 2009).

The corresponding statistics show only minor differences between the three processing procedures, indicating that our validation provides no definite outcome in favour for choosing a specific high-pass filter. The lowest mean absolute error (MAE) and highest fraction of explained variance were obtained for $CO_2$ fluxes calculated by linear detrending, followed by recursive filtering, while the lowest MAE and highest fraction of explained variance for $H_2O$ fluxes were observed for recursive filtering.

**4. Conclusion**

Even though the number of published eddy covariance COS flux measurements has increased significantly during the past few years (Asaf et al., 2013, Billesbach et al., 2014, Maseyk et al., 2014, Commane et al., 2015), this is the first study to



systematically examine the use of QCLAS instruments for making defensible COS flux measurements, the necessary processing steps and QA/QC procedures and characterise the random flux uncertainty.

Consistent with earlier flux measurements of other scalars (e.g. $CH_4$, $N_2O$) based on laser spectroscopy, we found flux measurements to be affected by laser drift (e.g. Mammarella et al., 2010) and noise (e.g. Smeets et al., 2009). In order to account for laser drift we used two high-pass filtering methods, linear detrending and recursive filtering, and compared the results against flux calculations based on simple block averaging. While the independent validation against IRGA-based $CO_2$ and $H_2O$ fluxes suggested minor differences between these three approaches, linear detrending and in particular recursive filtering clearly increased the fraction of data passing the various quality control criteria. We thus strongly advocate the use of Allan variance analysis for QCLAS-based COS flux measurements to quantify if, and from which integration time on, laser drift affects the calculation of covariances and thus fluxes. Likewise, consistent with other studies (e.g. Eugster et al., 2007) we found high frequency COS measurements to be affected by noise. As this noise appeared to be not entirely random and affected the high-frequency component of fluxes, we decided to filter these unwanted contributions (Smeets et al. 2009; Pierce et al., 2015). The flux underestimation due to the application of these digital low- and high-pass filters and additional filtering due to the imperfect sensor response, tube attenuation, etc. was corrected using a transfer function approach. The independent validation against IRGA-based $CO_2$ and $H_2O$ fluxes suggests that $CO_2$ and $H_2O$ fluxes calculated this way are comparable with the more established IRGA technology.

We conclude that the employed QCLAS can be used for defensible COS flux measurements, provided the appropriate corrections are made, with preference towards the use of recursive high-pass filtering.

### Acknowledgements

This work was funded by the Austrian National Science Fund (FWF) under contract P27176-B16. We give thanks to the family Hofer (Neustift, Austria) for kindly granting us access to the study site.

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





**Table 1** List of flags, their limit values and the percentage of data left after application, for each gas. First three criterions do not relate to the flux directly and hence, are unaffected by data processing and pertain to all gases and processing scenarios (block averaging (BA), linear detrending (LD), recursive filtering (RF)).

| Exclusion criteria | Limiting value | Data left (%) | | |
|---|---|---|---|---|
| Third rotation angle | $\leq 10\,°$ | 81 | | |
| Integral turbulence test | $\leq 100\,\%$ | 95 | | |
| Footprint | $\geq 80\,\%$ | 61 | | |
| COS | | BA | LD | RF |
| Flux range | -60 to 40 pmol m$^{-2}$ s$^{-1}$ | 86 | 90 | 95 |
| $u_*$ | $\leq 0.16$ m$^{-2}$ s$^{-1}$ (BA/LD) / 0.17 m$^{-2}$ s$^{-1}$1 (RF) | 46 | 46 | 43 |
| Signal to noise ratio | $\geq 1.96\,\sigma$ | 50 | 54 | 68 |
| Stationarity | $\leq 100\,\%$ | 72 | 78 | 84 |
| All exclusion criteria | - | 10 | 12 | 15 |
| CO$_2$ | | | | |
| Flux range | -30 to 30 µmol m$^{-2}$ s$^{-1}$ | 92 | 94 | 98 |
| $u_*$ | $\leq 0.12$ m$^{-2}$ s$^{-1}$ (BA/LD) / 0.13 m$^{-2}$ s$^{-1}$ (RF) | 60 | 60 | 57 |
| Signal to noise ratio | $\geq 1.96\,\sigma$ | 63 | 67 | 79 |
| Stationarity | $\leq 100\,\%$ | 80 | 84 | 89 |
| All exclusion criteria | - | 21 | 23 | 24 |
| H$_2$O | | | | |
| Flux range | -1 to 6 mmol m$^{-2}$ s$^{-1}$ | 91 | 92 | 94 |
| $u_*$ | - | - | - | - |
| Signal to noise ratio | $\geq 1.96\,\sigma$ | 54 | 58 | 66 |
| Stationarity | $\leq 100\,\%$ | 77 | 81 | 82 |
| All exclusion criteria | - | 25 | 27 | 31 |

30



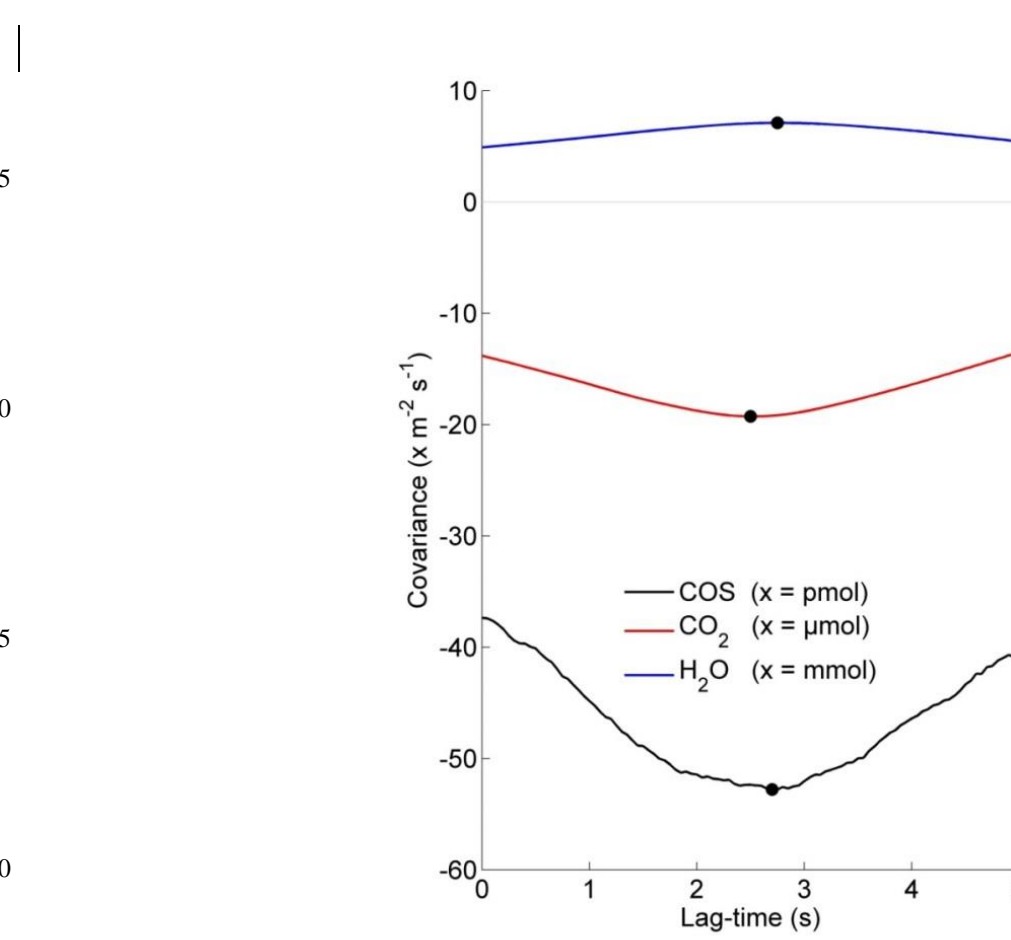

**Figure 1** Result of the cross-correlation function for the (mostly) tube-induced time delay among the gas signals and the vertical wind component. Lag times in this example were: COS = 2.6 s, $CO_2$ = 2.5 s, $H_2O$ = 2.85 s. Data correspond to 05.07.2015 10:00 am (CET).



**Figure 2** Plots of the Allan variance as a function of the averaging time τ as described in Werle (2010). The time series was obtained under ambient conditions in the instrument hut in the field by providing pressurized air from a cylinder to the QCLAS.





**Figure 3** Carbonyl sulphide cospectra (upper panels) and power spectra (lower panels) before (left panels) and after (right panels) application of the FIR filter. Solid and dashed lines in the cospectral plots indicate the reference model (Wohlfahrt et al., 2005a) and the attenuated reference model, the integrated ratio of which yields the low-pass filtering correction factor (1.15 in this case). Data correspond to 04.07.2015 08:15 am (CET).





**Figure 4** Allocation of BA, LD and RF filtered flux data for the stationarity test, represented for COS (upper panel), $CO_2$ (middle panel) and $H_2O$ (lower panel).





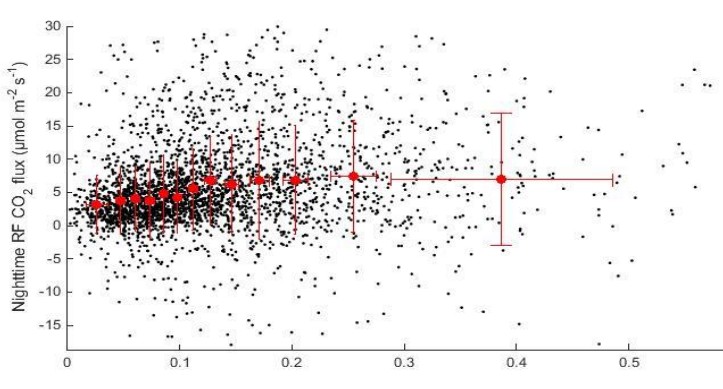

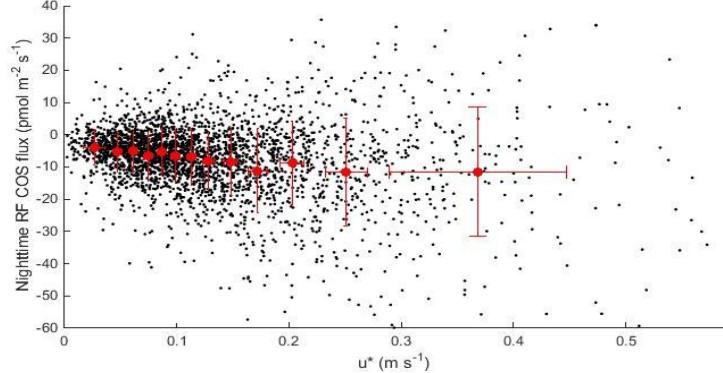

25 **Figure 5** Nocturnal RF filtered $CO_2$ (upper panel) and COS (lower panel) fluxes, restricted to their corresponding flux range ($CO_2$: -30–30 µmol m$^{-2}$ s$^{-1}$, COS: -60–40 pmol m$^{-2}$ s$^{-1}$), as a function of $u_*$. The flux data was divided into 13 bins, each bin comprising the same amount of data points (246 COS, 242 $CO_2$). For the RF filtered data series the $u_*$ thresholds were set to the 8th bin for $CO_2$ ($u_* = 0.13$ m s$^{-1}$) and the 10th bin for COS ($u_* = 0.17$ m s$^{-1}$).

30



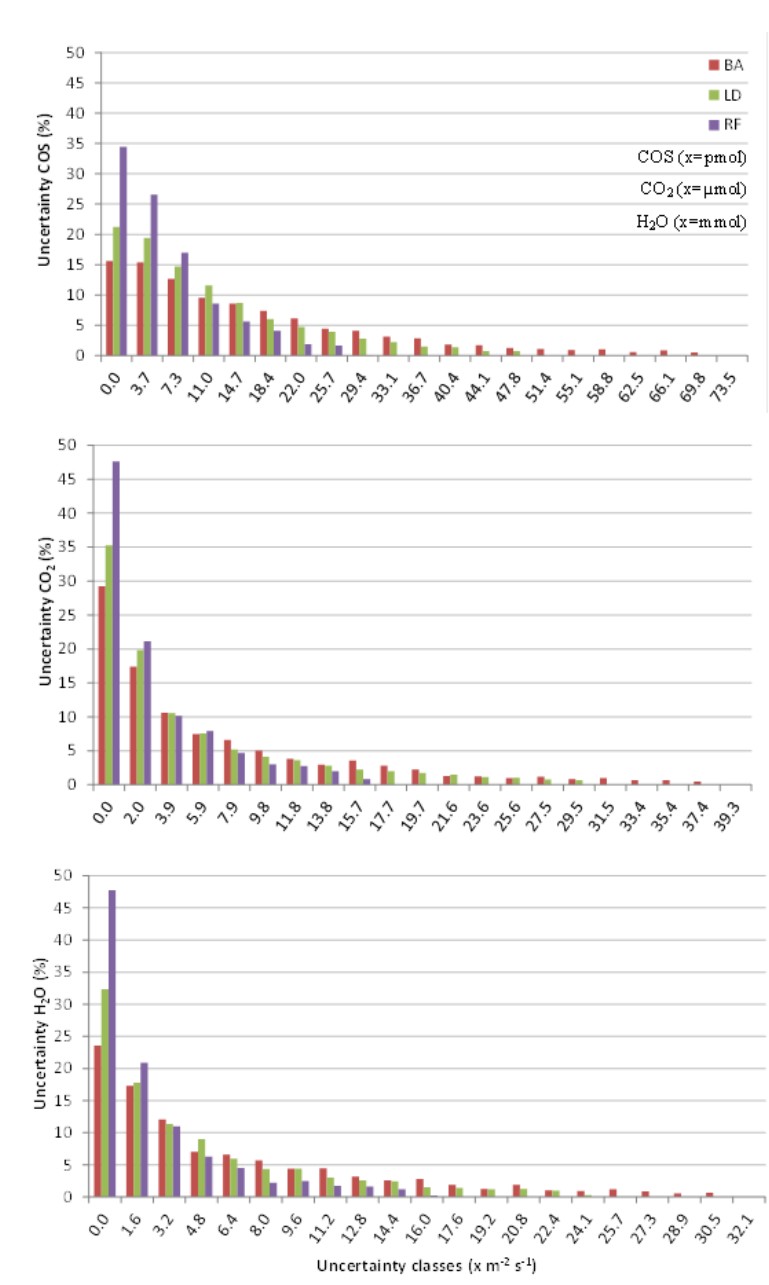

**Figure 6** Frequency distribution of calculated random uncertainty for different gases and different high-pass filtering scenarios.



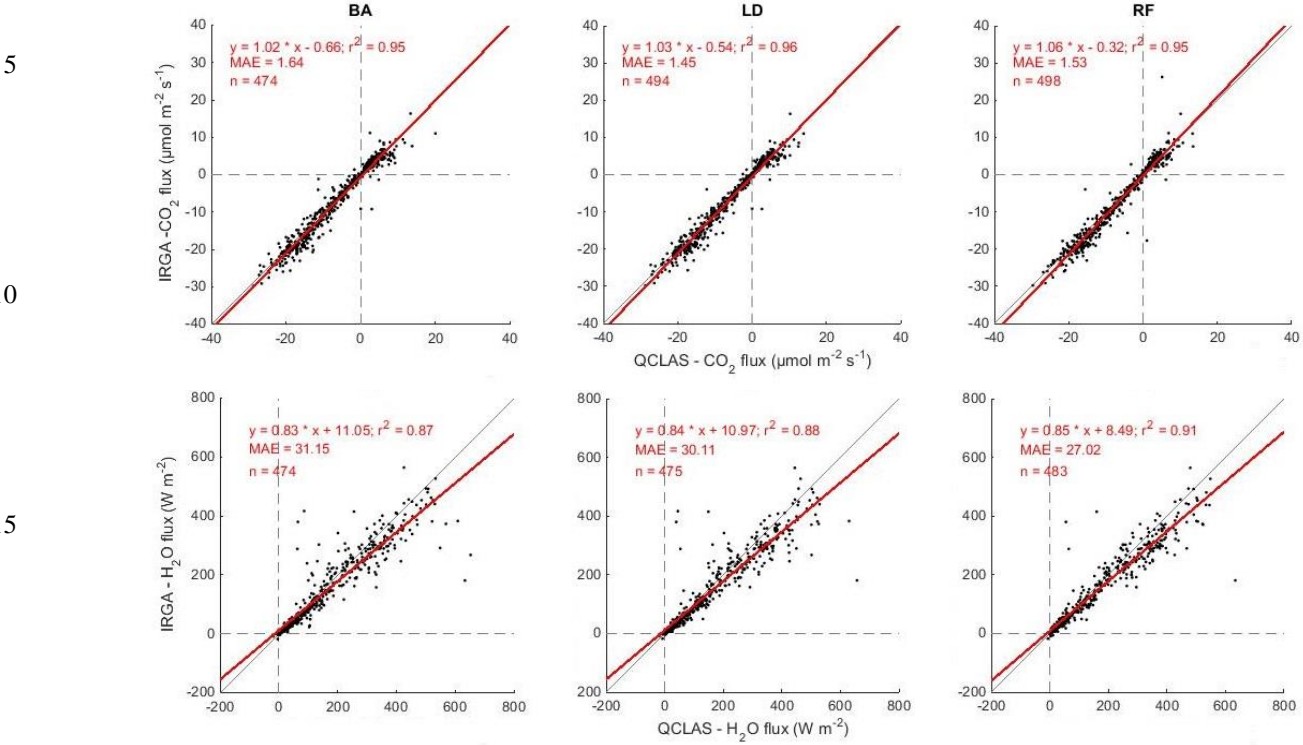

**Figure 7** Correlation between IRGA and QCLAS $CO_2$ (upper panels) and $H_2O$ (lower panels) fluxes using block averaging (left panels), linear detrending (middle panels) and a recursive filter with a 50 s time constant (right panels). The solid grey line indicates the 1:1 line and the solid red line is the geometric mean regression fit.