# Peer review of "Eddy covariance carbonyl sulphide flux measurements with a quantum cascade laser absorption spectrometer"

_Atmospheric Measurement Techniques, 2016_

## Referee Comment (RC1) · Anonymous Referee #1 · 2 Dec 2016

General Comments

Gerdel et al. present a careful consideration of eddy covariance data processing in the specific case of carbonyl sulfide (OCS) data from a popular, commercially available instrument. It is important to pay attention to this kind of methodological detail, and I respect the authors' work here. However, as to the manuscript itself, I have to say that I do not think it presents substantial new concepts, ideas, methods, or data, and therefore I cannot recommend it for publication in AMT. Please do not take the following explanation as demeaning the authors or their work; that is not my intent. I just need to be clear about why I would reject this manuscript.

The manuscript aims to assess whether the instrument in question can make "defen-

sible EC COS flux measurements", but what does "defensible" mean? The authors do not report the accuracy of the EC OCS fluxes, and so I came away from the manuscript with no more or less confidence in them than I had before. The validation (by comparison to another type of instrument) is restricted to $CO_2$ and $H_2O$ measurements. Instead of considering EC OCS accuracy, the authors focus on random noise, and the real question that the manuscript addresses is: can the noise in EC OCS measurements with this instrument be reduced via high- and low-pass filtering? The answer is of course yes: filtering out noise makes data less noisy. If the noise filtering techniques were new and innovative, their performance might be worth reporting, but as the manuscript itself says, the techniques are common.

The closest the authors come to the issue of bias (and therefore, in my mind, defensibility) is when they imply briefly in Section 3.3 that the high-frequency noise in the OCS mixing ratio tended to be correlated with the vertical wind velocity signal such that the cospectra were biased high. But they do not offer any evidence for this surprising claim, or any ideas for how such a thing might occur. The figures do not illustrate it.

The characterization of the noise in the OCS mixing ratio reported by the instrument (i.e. the Allan plots in Fig 2) does not appear to be new either; I believe the manufacturer itself has done this kind of analysis and freely shares it. That the characterization was done in the field here might add some novelty, but the authors do not say whether the noise was different in the field than in the manufacturer's labs.

This work seems to belong in the methods section of an article that actually makes use of the calculated EC OCS fluxes. In that case, details of how the data processing was done would be important to enable others to reproduce the work. I hope the authors have such a manuscript in the pipeline; I would look forward to reading it.

Specific Comments

I offer the following additional comments in the hopes that the authors might find them useful.

- I was a little concerned about how all laser spectrometers (for CO2 isotopes, N2O, CH4, OCS) were lumped together at times (e.g. page 8, lines 15-17), as if the present analysis ought to apply to all of them – but not, say, to an IRGA. What matters here is the noise and drift in the mixing ratio measurement, not whether the infrared light in the instrument came from a heated filament or a laser. The noise and drift considerations for Patrick Sturm's CO2 isotope QCLAS were very different than those for the present OCS QCLAS, which is measuring a comparatively tiny spectral line.

- Regarding the high-pass filtering: an alternate approach is to correct the drift in the OCS mixing ratio before beginning EC calculations. The OCS mixing ratio drift results from slow changes in the spectral baseline (i.e. the zero offset), which is reset periodically by the QCLAS's auto-background feature but changes in between those resets. By comparing measurements of the same gas (a standard tank or even the atmosphere) just before and just after an auto-background reset, one can determine how much the OCS zero level had drifted since the previous reset and make a linear correction (though the drift might not always be so linear). When it comes to the EC fluxes, this method is probably similar in effect to using linear detrending but ought to be better because real trends in the OCS mixing ratio would not be filtered out.

- Regarding the approach of Wienhold et al. (1995) to quantify the SNR, it always seemed fundamentally flawed to me. If you look at a plot of covariance vs lag time, you typically see oscillatory patterns because the eddies are quasi-periodic structures. So the variability in the covariance as you scan the lag time is not merely noise: much of it results from the quasi-periodic nature of the signal. You should be able to test this by comparing the noise estimate obtained this way when the signal (i.e. the real eddy flux) is large (e.g. at midday) to when it is near zero (e.g. at night for H2O and OCS). If the noise seems to decrease as the signal decreases, then the method may be flawed in the way I suggest.

---

## Referee Comment (RC2) · Anonymous Referee #2 · 7 Dec 2016

This manuscript presents a methodological approach to calculate eddy covariance flux of carbonyl sulfide (COS). The topic is very actual and interesting, however I cannot recommend the current version of the manuscript for publication in AMT, because of the following major points:

- Performance of QCLAS gas analysers have been already evaluated in the past for other gases (CH4 and N2O). The authors use filtering and analysis approaches to deal with laser drift affecting the low frequency and random noise affecting the high frequency, which are already well know in the flux community.

- The authors did not report a detailed description of EC processing steps and corrections, which I would expected for this kind of technical paper. For instance, it is not

clear for me if the COS dry mole fraction (corrected also for spectroscopic effect) was used for calculating fluxes (as it should be). In the data acquisition chapter it was only mentioned that "molar densities were measured . . ..".

- One of the main conclusion of the study is that fluxes obtained with several filtering strategies are not differing so much. Moreover, the validation is performed for CO2 and H2O against independent measurements, and not for COS.

- I agree that the use of recursive high-pass filtering is the only approach to deal with laser drift, especially in case of very small fluxes. However, by using this strategy, the true signal may be also filtered out. I believe that optimization of the setup (e.g. QCLAS insulation, minimize variation of ambient temperature, etc..) is the first prerequisite for obtaining defensible measurements.

- Related to the estimations of flux random uncertainty, I would recommend to look at the comprehensive paper by Rannik et al. (2016).

Minor comments:

- pag.4 L16. How much is the sensor separation (in vertical and horizontal directions)?

- pag.4 L22. 3d coordinate rotation is not recommended. Instead, standard methods are the 2d or planar fit.

- pag.4 L30-34 How the authors have decided on these threshold values ? How many 30 min runs are included in each of these subsamples?

- pag.5L10 and Fig.1 Why the cross-covariance functions look so smooth? Is this because of low-pass filtering? Please explain.

- chapter 3.3 and fig.3. The noise at high frequency range of the COS cospectrum is something normal, considering the probably low signal-to-noise ratio of this dataset, and the fact that a single run cospectrum is shown. Instead, I would recommend doing this kind of analysis using ensemble average cospectra. I am sure that visually the

noise will be much less. How the cospectra of CO2 and H2O look like?

- pag.7 L.15. Were the ustar thresholds visually estimated? Or how was it done?

- pag8 L.9-10 Sorry to say, but this is a very dangerous statement, which gives a wrong message to the reader. The random uncertainty is intrinsically part of EC flux measurements, and the low frequency fluctuations are not necessarily due only to instrumental noise (laser drift), but can be also real.

Rannik, Ü., Peltola, O., and Mammarella, I.: Random uncertainties of flux measurements by the eddy covariance technique, Atmos. Meas. Tech., 9, 5163-5181, doi:10.5194/amt-9-5163-2016, 2016.

---

## Author Comment (AC1) · 20 Jan 2017

Reply to Anonymous Reviewer #1: We thank reviewer #1 for his/her critical comments to which we reply below in a point-by-point fashion.

Reviewer comment: Gerdel et al. present a careful consideration of eddy covariance data processing in the specific case of carbonyl sulfide (OCS) data from a popular, commercially available instrument. It is important to pay attention to this kind of methodological detail, and I respect the authors' work here. However, as to the manuscript itself, I have to say that I do not think it presents substantial new concepts, ideas, methods, or data, and therefore I cannot recommend it for publication in AMT. Please do not take the following explanation as demeaning the authors or their work;

[Figure]

that is not my intent. I just need to be clear about why I would reject this manuscript.

Author reply: Not surprisingly we disagree with the reviewer regarding the novelty of our manuscript. Eddy covariance COS flux measurements have become possible since a few years (as of this writing we are aware of a total of 4 published studies) – so far these studies have provided minimal methodological detail on the eddy covariance data processing, making it very hard to reproduce the work. While QCLAS instruments have been used previously for eddy covariance flux measurements of several scalars, it cannot be assumed a priori that experience gained, e.g. for N2O and CH4, holds for COS as well, due to differences in e.g. line strength (as mentioned by this reviewer below). At a recent COS workshop in Finland in September 2016, the lack of clear guidelines for making COS flux measurements and processing the resulting data was identified as a key gap for progress on understanding ecosystem-scale COS exchange. We thus believe there is the need for a study which carefully analyzes and compares the various possible post-processing options associated with eddy covariance COS flux measurements obtained by the Aerodyne QCLAS, which is the only instrument suitable for eddy covariance so far, and thus provides future studies with a reference – this is what motivated this study.

Reviewer comment: The manuscript aims to assess whether the instrument in question can make "defensible EC COS flux measurements", but what does "defensible" mean? The authors do not report the accuracy of the EC OCS fluxes, and so I came away from the manuscript with no more or less confidence in them than I had before. The validation (by comparison to another type of instrument) is restricted to CO2 and H2O measurements. Instead of considering EC OCS accuracy, the authors focus on random noise, and the real question that the manuscript addresses is: can the noise in EC OCS measurements with this instrument be reduced via high- and low-pass filtering? The answer is of course yes: filtering out noise makes data less noisy. If the noise filtering techniques were new and innovative, their performance might be worth reporting, but as the manuscript itself says, the techniques are common.

Author reply: Accuracy is typically defined as the deviation of a measurement compared to some established standard or reference. For example, most of the COS community uses a gas standard derived from NOAA to reference COS concentration measurements. For eddy covariance flux measurements (of a scalar or vector quantity), such a standard or reference does not exist, and it is in fact difficult to imagine how such a standard would look like (this would require a reference surface with a known source/sink strength). As a consequence, the concept of accuracy does not apply to eddy covariance flux measurements. In contrast, it is common to quantify and report the systematic and random uncertainty (Moncrieff et al., 1996; GCB 2, 231-240), as for example in Sturm et al. (2012; AFM 152, 73-82). In the present manuscript we do not repeat previous analyses of systematic uncertainty due to e.g. the choice of the coordinate rotation, but instead focus on two sources of systematic uncertainty specific to the use of the QCLAS that is instrument drift, and the consequences that different approaches of high-pass filtering have, and high-frequency effects. The term "defensible" was meant to convey that we are not able to compare to an absolute standard, but instead have to quantify systematic uncertainty by exploring different processing options. We realize that this may have been not entirely clear and will explain in the revised manuscript what we mean by defensible through introducing the concept of systematic and random uncertainty following Moncrieff et al. (1996).

Reviewer comment: The closest the authors come to the issue of bias (and therefore, in my mind, defensibility) is when they imply briefly in Section 3.3 that the high-frequency noise in the OCS mixing ratio tended to be correlated with the vertical wind velocity signal such that the cospectra were biased high. But they do not offer any evidence for this surprising claim, or any ideas for how such a thing might occur. The figures do not illustrate it.

Author reply: Based on a comment by reviewer #2 we have completed a comprehensive (co )spectral analysis; in contrast to what we had observed on single half-hourly (co-)spectra, it turns out that averaging (co-)spectra by bins of stability and wind speed,
removes the observed behavior, suggesting it to be random noise. We will revise the corresponding text and present new figures showing the results of the new (co-)spectral analysis, as shown in the reply to reviewer #2 (Fig. R2_1).

Reviewer comment: The characterization of the noise in the OCS mixing ratio reported by the instrument (i.e. the Allan plots in Fig 2) does not appear to be new either; I believe the manufacturer itself has done this kind of analysis and freely shares it. That the characterization was done in the field here might add some novelty, but the authors do not say whether the noise was different in the field than in the manufacturer's labs.

Author reply: We thank the reviewer for this suggestion and we will certainly extend the paper by adding information about possible differences between the Allan variance plots reported by the manufacturer and our field findings.

Reviewer comment: This work seems to belong in the methods section of an article that actually makes use of the calculated EC OCS fluxes. In that case, details of how the data processing was done would be important to enable others to reproduce the work. I hope the authors have such a manuscript in the pipeline; I would look forward to reading it.

Author reply: What the reviewer suggests is the approach that has been taken in the few available COS flux papers so far, resulting in exactly the opposite effect – it is close to impossible to reproduce the published work based on these (necessarily within the frame of a paper not having a methodological focus) short descriptions. In addition, within a paper focused on "the science of COS fluxes", it would be impossible to present the details on different processing options and their consequences for QA/QC as present it here. We thus believe that there is a need for and value in such a detailed study in order to pave the way for future work focusing on the science. We though realize that we have missed out to demonstrate the consequences of the various processing options we have explored for the actual application of COS as a tracer for canopy photosynthesis and stomatal conductance and will add corresponding material

to the revised manuscript, as shown in Figure R1_1 below, which demonstrates the relationship between the ecosystem relative uptake rate (ERU – the ratio between the $CO_2$ to COS deposition velocity) as a function of incident photosynthetically active radiation. The figure demonstrates that the ERU deviates from a constant value at low PAR values, when the uptake of $CO_2$ decreases faster than COS due to biochemical limitations of photosynthesis. Important within the context of the manuscript is that the three high-pass filtering options all capture the same response.

Reviewer comment: I was a little concerned about how all laser spectrometers (for $CO_2$ isotopes, $N_2O$, $CH_4$, OCS) were lumped together at times (e.g. page 8, lines 15-17), as if the present analysis ought to apply to all of them – but not, say, to an IRGA. What matters here is the noise and drift in the mixing ratio measurement, not whether the infrared light in the instrument came from a heated filament or a laser. The noise and drift considerations for Patrick Sturm's $CO_2$ isotope QCLAS were very different than those for the present OCS QCLAS, which is measuring a comparatively tiny spectral line.

Author reply: Actually, as we argue above, it is exactly NOT our intention to lump all laser spectrometers together, rather this paper starts out to investigate whether the instrument used and the required processing steps are in any way different from what is known for other laser spectrometers. We will clarify this in the revised manuscript.

Reviewer comment: Regarding the high-pass filtering: an alternate approach is to correct the drift in the OCS mixing ratio before beginning EC calculations. The OCS mixing ratio drift results from slow changes in the spectral baseline (i.e. the zero offset), which is reset periodically by the QCLAS's auto-background feature but changes in between those resets. By comparing measurements of the same gas (a standard tank or even the atmosphere) just before and just after an auto-background reset, one can determine how much the OCS zero level had drifted since the previous reset and make a linear correction (though the drift might not always be so linear). When it comes to the EC fluxes, this method is probably similar in effect to using linear detrending but

ought to be better because real trends in the OCS mixing ratio would not be filtered out.

Author reply: We thank the reviewer for this useful suggestion which we have implemented thanks to the conducted half-hourly background measurements. The results of this approach will be compared against the high-pass filtering approaches already tested.

Reviewer comment: Regarding the approach of Wienhold et al. (1995) to quantify the SNR, it always seemed fundamentally flawed to me. If you look at a plot of covariance vs lag time, you typically see oscillatory patterns because the eddies are quasi-periodic structures. So the variability in the covariance as you scan the lag time is not merely noise: much of it results from the quasi-periodic nature of the signal. You should be able to test this by comparing the noise estimate obtained this way when the signal (i.e. the real eddy flux) is large (e.g. at midday) to when it is near zero (e.g. at night for H2O and OCS). If the noise seems to decrease as the signal decreases, then the method may be flawed in the way I suggest.

Author reply: The reviewer is correct in pointing out that cross-correlation plots feature repeating patterns due to the quasi-periodic structure of the main transporting eddies. The term SNR is thus not really appropriate and we rather should call it a flux detection limit test (as originally done by Wienhold et al. 1995). Based on the reviewer comment we have adopted a second flux detection limit test not based on a cross-correlation analysis – the approach by Pihlatie et al. (2005; BG 2, 377-387) and compare it against an improved version of the Wienhold approach put forward by Rannik et al. (2016; AMT 9, 5163-5181) as suggested by reviewer #2.
* * *
[Figure]

**Figure R1_1** Ecosystem relative uptake rate as a function of incident photosynthetically active radiation for three high-pass filtering options. Lines represent non-linear fits to the data (symbols) using the equation y = a / (b x).

**Fig. 1.**

---

## Author Comment (AC2) · 20 Jan 2017

Reply to Anonymous Reviewer #2: We thank reviewer #2 for his/her critical comments to which we reply below in a point-by-point fashion.

Reviewer comment: This manuscript presents a methodological approach to calculate eddy covariance flux of carbonyl sulfide (COS). The topic is very actual and interesting, however I cannot recommend the current version of the manuscript for publication in AMT, because of the following major points: - Performance of QCLAS gas analysers have been already evaluated in the past for other gases (CH4 and N2O). The authors use filtering and analysis approaches to deal with laser drift affecting the low frequency and random noise affecting the high frequency, which are already well know in the flux

**community.**

Author reply: While we agree that similar analyses have been carried out previously for QCLAS instruments measuring other trace gases (which in fact is discussed in the manuscript), we still believe there is the need to assess these corrections for COS and this particular QCLAS. This is particularly so as the few available COS eddy covariance flux publications focused on the "science" and thus necessarily provided little methodological detail. In particular none of the available COS flux papers did explore different processing options and their effects on QA/QC.

Reviewer comment: - The authors did not report a detailed description of EC processing steps and corrections, which I would expected for this kind of technical paper. For instance, it is not clear for me if the COS dry mole fraction (corrected also for spectroscopic effect) was used for calculating fluxes (as it should be). In the data acquisition chapter it was only mentioned that "molar densities were measured : : :.".

Author reply: We apologize for this omission – indeed we used dry mole fractions for calculating eddy covariance fluxes (superseding the need for the WPL density correction) – will be corrected in the revised manuscript.

Reviewer comment: One of the main conclusion of the study is that fluxes obtained with several filtering strategies are not differing so much. Moreover, the validation is performed for CO2 and H2O against independent measurements, and not for COS.

Author reply: As the author is likely aware of, the QCLAS used in this study is presently the only instrument providing the time response required for eddy covariance flux measurements – there is thus no possibility to directly validate the COS fluxes with a comparable method.

Reviewer comment: I agree that the use of recursive high-pass filtering is the only approach to deal with laser drift, especially in case of very small fluxes. However, by using this strategy, the true signal may be also filtered out. I believe that optimization
of the setup (e.g. QCLAS insulation, minimize variation of ambient temperature, etc..) is the first prerequisite for obtaining defensible measurements.

Author reply: Agreed, will make this point in the revised manuscript.

Reviewer comment: Related to the estimations of flux random uncertainty, I would recommend to look at the comprehensive paper by Rannik et al. (2016).

Author reply: Thanks for pointing us to this paper, which was published in AMT after we had submitted our manuscript. In response to this comment we have adopted the revised Wienhold-approach recommended by Rannik et al. to quantify the flux detection limit in our revised manuscript. In addition, based on a comment by reviewer #1, we have included the flux detection limit by Pihlatie et al. (2005; BG 2, 377-387).

Minor comments: Reviewer comment: pag.4 L16. How much is the sensor separation (in vertical and horizontal directions)?

Author reply: Horizontal sensor separation is 0.1 m perpendicular to the main wind direction, vertical sensor separation 0.1 m – this information will be added to the revised manuscript.

Reviewer comment: pag.4 L22. 3d coordinate rotation is not recommended. Instead, standard methods are the 2d or planar fit.

Author reply: While this is correct in principle, it is also well known that filtering for the third rotation angle (restricting to +/-10°) largely avoids issues with the 3D rotation – CO2 fluxes calculated with the 2D and 3D rotation agree to each other closely at this site, with a slope of 0.995, an offset of 0.128  $\mu$ mol m-2 s-1 and an R2 of 0.94. In fact, filtering for the third rotation angle removes extreme outliers that would otherwise pass.

Reviewer comment: pag.4 L30-34 How the authors have decided on these threshold values ? How many 30 min runs are included in each of these subsamples?

Author reply: The bin sizes for the determination of the random flux uncertainty follow-
ing Hollinger and Richardson (2005; TP 25, 873-885) were determined by Wohlfahrt et al. (2008; JGR, 10.1029/2007JD009286) for this site and are similar to those originally proposed by Hollinger and Richardson (2005) – will add this information to revised manuscript; based on these criteria, "similar" environmental conditions are identified during subsequent days and the difference in fluxes used as a measure of random uncertainty – so there are no subsamples created.

Reviewer comment: pag.5L10 and Fig.1 Why the cross-covariance functions look so smooth? Is this because of low-pass filtering? Please explain.

Author reply: This, admittedly very neat, example was calculated by block-averaging, so no additional filtering was applied.

Reviewer comment: chapter 3.3 and fig.3. The noise at high frequency range of the COS cospectrum is something normal, considering the probably low signal-to-noise ratio of this dataset, and the fact that a single run cospectrum is shown. Instead, I would recommend doing this kind of analysis using ensemble average cospectra. I am sure that visually the noise will be much less. How the cospectra of CO2 and H2O look like?

Author reply: Based on this reviewer comment we have conducted a comprehensive (co-)spectral analysis averaging data by bins of stability and wind speed; the analysis indeed shows that the erratic behavior at higher frequencies (noise) vanishes when averaged over a large enough sample; we will show the results for COS, CO2 and H2O in a new figure (Figure R2\_1 shown below) replacing former Figure 3; as a by-product of the cospectral analysis we have adopted the low-pass filtering correction procedure by Aubinet et al. (2001; AFM 108, 293-315) – the corresponding results will be presented in an additional figure

Reviewer comment: pag.7 L.15. Were the ustar thresholds visually estimated? Or how was it done?
Author reply: Ustar thresholds were determined visually – in the revised paper we will report ustar thresholds based on the change point detection algorithm by Barr et al. (2013; AFM 171-172, 31-45); as shown below (Fig. R2\_2), the new objective procedure has changed the ustar thresholds somewhat.

Reviewer comment: pag8 L.9-10 Sorry to say, but this is a very dangerous statement, which gives a wrong message to the reader. The random uncertainty is intrinsically part of EC flux measurements, and the low frequency fluctuations are not necessarily due only to instrumental noise (laser drift), but can be also real.

Author reply: This is just a factual statement reporting that high-pass filtering reduces the random uncertainty – but we understand the reviewer's concern and will modify the sentence correspondingly.

**AMTD**
**AMTD**

Figure R2\_1 Spectra (upper panels), cospectra (middle panels) and experimental transfer function (lower panels) for COS, CO2 and H2O. Solid lines and shaded areas refer to the average and one standard error. Data have been filtered for unstable conditions and wind speeds between 1.00 and 1.25 m s-1. The dashed line in the upper panel indicates the expected -5/3 decay in the inertial subrange; vertical dotted lines in the middle panel encompass the frequencies between which cospectra were normalized to each other; vertical dotted lines in the lower panels indicate the halfpoint frequency. Only data indicated by closed symbols in the lower panels were used to estimate the half-point frequency.
Fig. 1.
Figure R2\_2 U\* thresholds (red vertical lines) determined with the change-point-detection after Barr et al. (2013).

Fig. 2.